# Pro-Inflammatory Cytokines Trigger the Overexpression of Tumour-Related Splice Variant RAC1B in Polarized Colorectal Cells

**DOI:** 10.3390/cancers14061393

**Published:** 2022-03-09

**Authors:** Joana F. S. Pereira, Cláudia Bessa, Paulo Matos, Peter Jordan

**Affiliations:** 1Department of Human Genetics, National Institute of Health ‘Dr. Ricardo Jorge’, 1649-016 Lisbon, Portugal; joana.pereira@insa.min-saude.pt (J.F.S.P.); claudia.bessa@insa.min-saude.pt (C.B.); paulo.matos@insa.min-saude.pt (P.M.); 2BioISI—Biosystems & Integrative Sciences Institute, Faculty of Sciences, University of Lisbon, 1749-016 Lisbon, Portugal

**Keywords:** colorectal cancer, inflammation, interleukin, macrophage, RAC1, signal transduction

## Abstract

**Simple Summary:**

Tumours are now known to develop more quickly when the tumour cell mass is located in a tissue that shows signs of chronic inflammation. Under such conditions, inflammatory cells from the surrounding tumour microenvironment provide survival signals to which cancer cells respond. We have previously found that some colorectal tumours overexpress the protein RAC1B that sustains tumour cell survival. Here we used a colon mucosa-like in vitro cell model and found that the presence of cancer-associated fibroblasts and pro-inflammatory macrophages stimulated colorectal cells to increase their RAC1B levels. Under these conditions, the secreted survival signals were analysed, and interleukin-6 identified as the main trigger for the increase in RAC1B levels. The results contribute to understand the tumour-promoting effect of inflammation at the molecular level.

**Abstract:**

An inflammatory microenvironment is a tumour-promoting condition that provides survival signals to which cancer cells respond with gene expression changes. One example is the alternative splicing variant Rat Sarcoma Viral Oncogene Homolog (Ras)-Related C3 Botulinum Toxin Substrate 1 (RAC1)B, which we previously identified in a subset of V-Raf Murine Sarcoma Viral Oncogene Homolog B (BRAF)-mutated colorectal tumours. RAC1B was also increased in samples from inflammatory bowel disease patients or in an acute colitis mouse model. Here, we used an epithelial-like layer of polarized Caco-2 or T84 colorectal cancer (CRC) cells in co-culture with fibroblasts, monocytes or macrophages and analysed the effect on RAC1B expression in the CRC cells by RT-PCR, Western blot and confocal fluorescence microscopy. We found that the presence of cancer-associated fibroblasts and M1 macrophages induced the most significant increase in RAC1B levels in the polarized CRC cells, accompanied by a progressive loss of epithelial organization. Under these conditions, we identified interleukin (IL)-6 as the main trigger for the increase in RAC1B levels, associated with Signal Transducer and Activator of Transcription (STAT)3 activation. IL-6 neutralization by a specific antibody abrogated both RAC1B overexpression and STAT3 phosphorylation in polarized CRC cells. Our data identify that pro-inflammatory extracellular signals from stromal cells can trigger the overexpression of tumour-related RAC1B in polarized CRC cells. The results will help to understand the tumour-promoting effect of inflammation and identify novel therapeutic strategies.

## 1. Introduction

The mucosa of our gastrointestinal tract forms a physiological barrier that separates the surrounding tissues from the gut lumen, containing ingested food as well as microorganisms and environmental toxins [1]. This intestinal barrier mostly relies on a single epithelial cell layer, composed of different cell types, and a mucus layer. In particular, highly polarized columnar cells form a tight-junction belt that seals the intercellular spaces and restricts any uptake of molecules or particles to highly selective transport mechanisms at the apical surface. This polarized cell layer is supported in the underlying submucosal tissue by a high immune cell surveillance.

Colorectal cancer is one of the leading causes of death [2] and tumour formation generally initiates with an oncogenic mutational event in a colon mucosa cell. For example, activating mutations in the Kirsten Rat Sarcoma Viral Oncogene Homolog (*KRAS*) or V-Raf Murine Sarcoma Viral Oncogene Homolog B (*BRAF*) oncogenes confer proliferative characteristics and are found in 30% or 15% of sporadic colorectal tumours, respectively [3,4,5]. For malignant progression to proceed, a pro-inflammatory microenvironment has been recognized as a crucial tumour-promoting condition [6,7]. For example, patients with inflammatory bowel diseases (IBDs) have increased risk of developing colorectal cancer (CRC) [8,9,10,11]. 

A major molecular connection between cancer and inflammation resides in chemokines, soluble factors released by immune cells to regulate their response, which also stimulate the proliferation or survival of most human neoplastic cell types, including colorectal cells [12,13,14,15,16]. In contrast, the genetic alterations selected in cancer cells lead to the secretion of factors that stimulate stromal cells and propagate the inflammatory microenvironment. 

Within the tumour microenvironment (TME), stromal fibroblasts and macrophages play particularly important roles. Fibroblasts are the main cell type in our connective tissues and do not form ordered cell layers but remain as quiescent individual cells in the extracellular matrix (ECM). They become activated by tissue damage or inflammation and then proliferate. Similar to tissue repair, tumour development is also associated with fibroblast activation, then designated as tumour- or cancer-associated fibroblasts (CAFs). CAFs secrete ECM remodelling enzymes as well as cytokines for the recruitment of immune cells, including macrophages. Macrophages are immune cells known for their phagocytic capacity and are recruited in response to inflammatory conditions [17,18]. Different macrophage populations have been recognized. M1 macrophages secrete pro-inflammatory mediators, which are required to eliminate pathogens or damaged cells. They can be activated in vitro by interferon-gamma (IFN-γ) and lipopolysaccharide (LPS). M2 macrophages, on the other hand, are activated in vitro by interleukin (IL)-4 and secrete anti-inflammatory cytokines to downregulate the inflammatory process and allow for wound-healing and tissue repair [19,20]. Besides the traditional M1 and M2-type macrophages, a wider spectrum of macrophage differentiation states is currently under investigation, revealing intrinsic plasticity and mutual regulation [21,22].

Soluble cytokines from stromal fibroblasts or macrophages alter the behaviour of colorectal cells by different mechanisms, including transcriptional activation, an altered ECM composition and metabolic cooperation [18,23,24]. Previously, we described that changes in alternative splicing constitute a further layer, through which inflammatory conditions can act on colon cells. We identified an increased expression of an alternative splicing variant of the small GTPase Ras-Related C3 Botulinum Toxin Substrate 1 (RAC1) [25]. Variant RAC1B is also overexpressed in a subtype of colorectal tumours, characterized by the presence of mutant *BRAF* and corresponding to up to 15% of all sporadic CRCs [26,27]. Splicing variant RAC1B contains additional 19 amino acids that modify its functional properties, such as adopting a predominantly active, GTP-bound conformation and preferentially stimulating the pathway that leads to activation of transcription factor Nuclear Factor of Kappa Light Polypeptide Gene Enhancer In B-Cells (NF-κB) [28,29,30]. These properties act synergistically with the BRAF-V600E mutation to sustain cell survival [26]. Our studies further suggested that the overexpression of RAC1B is a subsequent genetic event in tumour-initiated colorectal cells, allowing them to escape from oncogene-induced senescence [31] and continue malignant progression. However, mutations in the RAC1B gene that could explain a shift from RAC1 to RAC1B transcripts have not been found, so the mechanism involved in triggering RAC1B overexpression remains to be determined. 

Here, we identified soluble pro-inflammatory cytokines that triggered RAC1B overexpression. For this, we analysed colorectal Caco-2 and T84 cells, grown as a polarized epithelial-like cell layer on filter membranes and sharing their basolateral medium with that of CAFs and/or macrophage subtypes. 

## 2. Materials and Methods

### 2.1. Cell Culture 

Caco-2, THP-1, NCM460 and HT29 cells were maintained in Roswell Park Memorial Institute 1640 Medium (RPMI), while DLD-1 and T84 colorectal cells were cultured in Dulbecco’s modified Eagle medium (DMEM) or DMEM/nutrient mixture F- 12 (DMEM/F-12), respectively. All were supplemented with 10% (*v*/*v*) of heat inactivated foetal bovine serum (FBS) (cell line characteristics are summarized in Appendix A). NIH3T3 cells were maintained in DMEM supplemented with 10% (*v*/*v*) of heat inactivated new-born calf serum (NBCS). CT5.3 cancer-associated fibroblasts (immortalized cell line established from a resected colorectal adenocarcinoma [32]; gift from O. Wever, Ghent, Belgium) were grown in DMEM supplemented with 10% (*v*/*v*) FBS, 2.5 μg/mL puromycin and 100 U/mL penicillin/streptomycin (PEN/STREP) (all reagents from Thermo Fisher Scientific, Waltham, MA, USA). Cells were maintained at 37 °C with 5% CO_2_, and regularly checked for an absence of mycoplasma infection by PCR amplification of a 16S ribosomal DNA fragment [primers forward (F) 5′ ACTCCTACGGGAGGCAGCAGTA 3′ and reverse (R) 5′ TGCACCATCTGTCACTCTGTTAACCTC 3′] from lysates of cells harvested from the culture medium. 

### 2.2. THP-1 Cell Differentiation 

THP-1 monocyte cells were differentiated with 50 ng/mL phorbol-12-myristate 13-acetate (PMA) (Sigma-Aldrich, St. Louis, MO, USA) for 24 h into a M0 macrophage phenotype, with cell adhesion and spreading evaluated under an optical microscope. After this, M0 cells were re-fed for 24 h with fresh medium containing either 10 ng/mL lipopolysaccharides (LPS, L4516 Sigma-Aldrich, St. Louis, MO, USA) and 10 ng/mL interferon (IFN)-γ (Gibco, Thermo Fisher Scientific, Waltham, MA, USA), or 10 ng/mL interleukin (IL)- 4 (R&D systems) to differentiate into M1 and M2 macrophages, respectively. Successful differentiation was evaluated by PCR amplification of previously described marker genes (see Appendix A).

### 2.3. Cell Polarization and Co-Culture Assays 

For cell polarization, Caco-2 cells were grown on porous (1 μm) transwell polyester (PET) filter inserts (24-well size, 6.4 mm diameter and 0.3 cm^2^ area, Corning) in RPMI medium supplemented with 5% (*v*/*v*) FBS for 10–15 days, until they reached a transepithelial electric resistance (TEER) of from 1000 to 1200 Ω, as measured with a chopstick electrode STX2 (World Precision Instruments, Sarasota, FL, USA) [33,34]. T84 cells were polarized in DMEM/F12 medium supplemented with 5% (*v*/*v*) FBS for 10–15 days, until they reached a TEER of from 1500 to 2000 Ω. For co-culture assays, epithelial and stromal cells were first grown in 24-well plates for 24 h, and then the filter inserts with polarized Caco-2 or T84 cells, or with non-polarized cells, were added. The co-cultures were maintained during different time periods (24 h, 48 h, 72 h, 96 h or 120 h). The conditioned basolateral media from co-cultures were recovered and frozen at −80 °C for later antibody array analysis. 

### 2.4. Cell Treatments

For treatment with purified cytokines, polarized Caco-2 or T84 cells were incubated from the basolateral side, for 48 h, with either the control vehicle phosphate-buffered saline (PBS) + 0.01% bovine serum albumin (BSA, Sigma-Aldrich, St. Louis, MO, USA) or with 1, 10, or 100 ng/mL of the following cytokines: IL-1β, IL-6, IL-11, granulocyte-macrophage colony-stimulating factor (GM-CSF) (all from R&D Systems, Minneapolis, MN, USA). The cytokine stock solutions were prepared at least 1000-fold concentrated in PBS + 0.01% BSA and stored at −20 °C. For cytokine neutralization, the control vehicle PBS or anti-human IL-6 or IL-1β antibodies (AF-206-SP or MAB601-SP, R&D Systems, Minneapolis, MN, USA) were added from a 1000-fold stock to a final concentration of 500 ng/mL or 1000 ng/mL for 48 h to the basolateral medium of polarized Caco-2 cells. 

### 2.5. Qualitative and Quantitative Reverse Transcription-Polymerase Chain Reaction (RT-PCR) 

Total RNA was extracted with the RNA isolation kit (Macharey-Nagel, Düren, Germany) and reverse transcribed using random primers (Thermo Fisher Scientific, Waltham, MA, USA) and Ready-to-Go You-Prime First Strand Beads (Cytiva, Marlborough, MA, USA). Qualitative amplification reactions from cells were performed using GoTaq G2 Flexi DNA polymerase (Promega, Madison, WI, USA), using the primers and annealing temperatures summarized in Appendix A for each marker gene. All reactions included an initial denaturation step of 5 min at 94 °C and a final extension step of 10 min at 72 °C. To allow for a semi-quantitative analysis of transcript levels, all amplification conditions were experimentally optimized to correspond to the linear amplification phase, using serial dilutions of control cDNAs. The products were separated on 2% agarose gels containing ethidium bromide and band intensities were quantified on digitalized images using ImageJ software (version 1.53f51, National Institutes of Health, Bethesda, MD, USA), followed by normalization to GAPDH expression levels. Quantitative RT-PCR (qRT-PCR) of RAC1 transcripts in colorectal cells was performed on an ABI Prism 7000 Sequence Detection System (Thermo Fisher Scientific, Waltham, MA, USA). Primers were designed using the ABI Primer Express software, which amplified amplicons specific to endogenous RAC1B (78 bp) (F: 5′ GGGCAAAGACAAGCCGATTG 3′ and R: 5′ CGGACATTTTCAAATGATGCAGG 3′) or total RAC1 transcripts [(RAC1 + RAC1B; 75 bp) (F: 5′ CCTGCATCATTTGAAAATGTCCG 3′ and R: 5′ CCCACTAGGATGATGGGAGTGT 3′)]. Each cDNA sample was diluted 5-fold to guarantee accurate pipetting and 5 μL was added to 300 nmol/L primers and SYBR Green Master Mix (Applied Biosystems, Thermo Fisher Scientific, Waltham, MA, USA). The cycling conditions comprised 10 min polymerase activation at 95 °C and 40 cycles at 95 °C for 15 s and 60 °C for 30 s. For standardization, all samples were analysed against the same control Caco-2 cDNA reference sample using the 7000 SDS 1.1 RQ Software (ΔΔ CT method). Each amplification was performed in triplicate reactions. All RT-PCR data were obtained from at least three independent experiments. 

### 2.6. Western Blot (WB) Procedures 

Cells were lysed in 50 μL of lysis buffer [50 mM Tris/HCl (pH 7.5), 2 mM MgCl_2_, 100 mM NaCl, 10% (*v*/*v*) glycerol, 1% (*v*/*v*) NP40] and total proteins from co-culture assays were separated in 10 or 12% (*w*/*v*) SDS-PAGE gels. Following electrophoresis, proteins were transferred onto a polyvinylidene difluoride (PVDF) membrane (Bio-Rad, Hercules, CA, USA). WB membranes were blocked in 5% (*w*/*v*) milk powder in wash buffer (TBS with 0.5% (*v*/*v*) Triton X-100) and specific proteins probed using the indicated primary antibodies overnight, followed by three wash steps and incubation with a goat anti-mouse or anti-rabbit IgG horseradish peroxidase (HRP) conjugate (170-6516, 170-6515; Bio-Rad, Hercules, CA, USA). Protein bands were visualized by chemiluminescence on X-ray films and quantified on digitalized images by densitometric analysis with ImageJ software (version 1.53f51, National Institutes of Health, Bethesda, MD, USA). Primary antibodies were: mouse anti-α-tubulin (T5168) from Sigma-Aldrich, St. Louis, MO, USA; mouse anti-E-cadherin (610181) from BD Biosciences, San Carlos, CA, USA; mouse anti-RAC1 (23A8, 05-389) and rabbit anti-RAC1B (09-271) from Merck Millipore, Darmstadt, Germany; rabbit anti-STAT3 (30835) and rabbit anti phospho-STAT3 (Y705) (D3A7) from Cell Signaling Technology, Danvers, MA, USA. All original WB film exposures used to assemble the Figures can be found in Appendix A.

### 2.7. Cytokine Array Procedure 

A total of 1 mL of conditioned basolateral medium from co-cultures was added to human inflammation antibody array C3 (AAH-INF-3-8; RayBiotech Life Inc., Peachtree Corners, GA, USA) membranes, which were previously blocked and processed according to the instructions of the manufacturer. The membranes were shaken overnight at 4 °C in separate wells of an 8-well incubation tray and then washed. A total of 1 mL of a 1:500 dilution of biotinylated antibody cocktail was added to each membrane, and the mixture was incubated on a shaker overnight at 4 °C. After washing, the membranes were incubated with 2 mL of 1× streptavidin-conjugated peroxidase for 2 h at room temperature (rt). Following thorough washing, the membranes were exposed to a peroxidase substrate (detection buffer mixture) for 1 min in the dark, before exposure to X-ray films for times ranging from 5 s to 5 min. X-rays films were digitalized, and images analysed with ImageJ software (version 1.53f51, National Institutes of Health, Bethesda, MD, USA). The 6 positive-control spots were used for normalization of signal intensities between arrays, the six negative-control spots to measure the nonspecific baseline signal, and blank spots (BLANK) to measure the background signal. For each spot, the remaining density grey level was obtained by subtracting the background grey levels from the total raw density grey level values. Then, the mean value was calculated from the duplicates of each spot. The relative fold-difference in each cytokine signal was then determined by dividing the mean value from the co-culture condition through the mean value from the control culture condition. 

### 2.8. Confocal Immunofluorescence Microscopy 

Cells were grown on PET transwell filters, washed twice in PBS, immediately fixed with 4% (*v*/*v*) formaldehyde in PBS for 20 min at rt, and subsequently permeabilized with 0.5% (*v*/*v*) Triton X-100 in PBS for 30 min at rt. Cells were then labelled for 2 h with primary antibodies (see above) against RAC1B (1:250), E-cadherin (1:200), or ZO-1 (1:250; sc-10804 from Santa Cruz Biotechnology, Santa Cruz, CA, USA), and washed 3× in PBST (PBS + 0.01% Tx-100) for 5 min with gentle shaking, followed by 30 min incubation with a 1:250 dilution of mouse or rabbit Alexa Fluor 488 or mouse Alexa Fluor 546 (Thermo Fisher Scientific, Waltham, MA, USA) and phalloidin-TRITC (Sigma-Aldrich, St. Louis, MO, USA). The anti-RAC1B antibody was previously validated for this technique [35,36]. Cells were washed 3× in PBS, briefly stained with 1.25 μg/mL DAPI (Sigma-Aldrich, St. Louis, MO, USA), washed again, post-fixed with 4% (*v*/*v*) formaldehyde in PBS for 10 min at rt. Then cover slips were mounted on glass slides in VectaShield (Vector Laboratories, Burlingame, CA, USA) and sealed with nail polish. The 405 nm, 488 nm and 532 nm laser lines of a Leica TCS-SPE confocal microscope were used to acquire one Airy thick XZ, and Z-stacks of XY images from polarized monolayers in transwell filters. Recorded images were processed with Leica in-built software and assembled in figures with Adobe Photoshop software (CS4, version 11.0). 

### 2.9. Statistical Analysis

Data were analysed using one-way or two-way ANOVA tests followed by the indicated post hoc tests, with *p* < 0.05 accepted as the statistical significance level. The data that are shown reflect the mean ± SEM from at least three independent experiments.

## 3. Results

### 3.1. Expression of RAC1B in Caco-2 Cells Is Modulated by Co-Cultured Stromal Cells

We tested the above-mentioned hypothesis that microenvironmental stimuli could be the trigger for changes in the expression of RAC1B in colorectal tumour cells. For this, a cell line capable of forming a polarized epithelial-like cell layer was used, and Caco-2 adenocarcinoma cells are the most widely used model to represent the organization and signalling in the epithelial barrier of the mucosa [37,38]. As shown in Figure 1A,B, Caco-2 cells grown on a microporous filter membranes formed a fully polarized cell layer with high transepithelial electrical resistance (TEER) and an apical actin belt. RAC1B was detected at the basolateral membranes. The layer contained ZO-1-positive tight junctions and increased E-cadherin expression (Appendix A). 

The expression of endogenous RAC1B protein was detected by Western blot (WB) in whole-cell lysates from polarized Caco-2 cells, as well as in non-polarized Caco-2. Al-though RAC1B can readily be detected in other colon cell lines (HT29, T84 or NCM460), it was virtually absent in stroma-derived cell types, including fibroblasts (NIH3T3, CT5.3) and monocytes (THP-1) (Figure 1C). Interestingly, we observed that RAC1B levels were lowest in normal NCM460 colonocytes [39] and in polarized Caco-2 cells, consistent with previous reports that related its overexpression to tumour progression [26,31,40,41].

Next, we determined whether RAC1B levels in the fully polarized Caco-2 cells would respond to the presence of stromal cell types. For this, polarized Caco-2 cells on filter inserts of 1 µm pore size were placed onto 24-well plates to share the culture medium with either pre-seeded NIH3T3 fibroblasts, THP-1 monocytes, or non-polarized Caco-2 cells (as negative control). Following co-culture times of 0 h, 24 h, 48 h, 72 h, and 96 h, the TEER of the upper polarized Caco-2 layer was determined and then cells were lysed to compare total RAC1B protein levels by WB. Figure 2 and Appendix A show that, under control conditions or in the presence of undifferentiated monocytes, the RAC1B levels dropped to roughly 50% as the Caco-2 layer aged for a period of 96 h, suggesting that RAC1B levels decreased with the structural differentiation of the epithelial-like monolayer. In contrast, the presence of fibroblasts in the basal medium promoted a sustained 1.5-fold increase in total RAC1B in Caco-2 cells for the first 48 h, before starting to drop as the culture aged. When the TEER values of the Caco-2 layer were compared, the fibroblast co-culture revealed a rapid decrease in resistance during the first 48 h (Figure 2A), which correlated with morphological alterations in the epithelial layer integrity detected by confocal microscopy (Figure 2C). The control co-culture with Caco-2 cells showed an initial increase in TEER until a stable value was reached after 48 h, while the presence of THP-1 cells led to a transient TEER decrease after 24 h, which then rapidly recovered and approached the control values. In contrast to the co-culture with fibroblasts, confocal microscopy analysis revealed no morphological changes under the latter two conditions.

From these preliminary studies, we concluded that a co-culture time of 48 h allows for the effects on RAC1B protein levels in the polarized Caco-2 layer to be observed, and selected this condition for further experiments. We then compared this co-culture condition and the effect of co-culturing the human CAF cell line CT5.3 [32], or THP-1-derived cells differentiated via M0 into M1-like or M2-like macrophages (Appendix A). In addition, triple co-cultures with the simultaneous presence of fibroblasts and macrophages were tested. These experiments showed that human CAFs or co-culture with M1-like, but not M2-like, macrophages, led to a clear increase in RAC1B expression in the polarized Caco-2 layer (Figure 3A). Finally, the combination of human CAFs with M1-like macrophages had the strongest and statistically most significant effect on increasing RAC1B protein in Caco-2 cells. Although we chose to monitor the RAC1B protein levels at 48 h of co-culture, the corresponding endogenous RAC1B transcript levels also increased, reaching a maximal response earlier, at 24 h (Figure 3B), which was compatible with the progressive decrease in protein levels observed at longer culture times (see Figure 2B).

### 3.2. Cytokines from the Co-Culture Medium Trigger the Changes in RAC1B Protein Levels

In our experimental setting, the co-cultured stromal cells were attached to the culture dish and did not physically communicate with the polarized Caco-2 layer grown on top of the filter insert; however, they shared, through the filter membrane’s pores, the same culture medium as the basolateral Caco-2 membrane domain. Thus, we next analysed which soluble factors could mediate the observed increase in RAC1B levels. All co-culture media were preserved and individually analysed with 40 cytokine-antibody arrays. The individual blot images were digitalized, the signal intensities were quantified, and values were normalized against the positive and negative control spots (Figure 4A). First, the direct pixel intensities were displayed (Figure 4B) and revealed the most pronounced changes in the levels of IL-6, IL-8, RANTES and MCP1. Then, signal intensities were normalized to the values obtained in the control co-culture with Caco-2 itself, to adjust for medium exhaustion and cytokine background levels (Figure 4C). This normalization revealed IL-6, IL-1β and GM-CSF as the main differences associated with increased RAC1B expression. As an unaffected cytokine, IL-11 was chosen for further control experiments (Figure 4B,C). 

To confirm the identified cytokines, filter membranes with polarized Caco-2 cell layers were cultured in dishes containing cell-free medium, and then the selected cytokines were added in purified form, instead of any co-cultured cell. As shown in Figure 5, the selected cytokines were added in three increasing concentrations and a dose-dependent increase in RAC1B levels was confirmed for IL-6 and IL-1β (Figure 5A,B). Under the same experimental conditions, no significant increase was observed with the array-derived candidate GM-CSF, or with the negative control IL-11 (Figure 5C,D). In addition, when a combination of recombinant IL-6 and Il-1β was tested, no synergistic or additive effect was observed on Caco-2 RAC1B levels (Figure 5E).

### 3.3. Interleukin-6 Is a Key Factor Stimulating RAC1B Expression in Polarized Caco-2 Cells

As a further validation step for the identified cytokines, neutralizing antibodies were tested for their ability to block the observed effect on RAC1B expression under co-culture conditions. For this, validated antibodies against human IL-6 or IL-1β were added to the basolateral medium during the co-culture with CAFs and M1 macrophages. We detected that the increase in RAC1B protein levels in Caco-2 cells was completely blocked by the addition of 500 ng/mL of the neutralizing anti-human IL-6 antibody, whereas anti-IL-1β antibodies were only partially effective (Figure 6A,B).

In these experiments, we also monitored the stimulation of STAT3 phosphorylation, which is a typical cell signalling response for IL-6; however, it is not typical for IL-1β. Indeed, when we analysed Caco-2 cell lysates after co-culture with CAFs and M1 macrophages, we observed an increase in the phosphorylation of STAT3 at Y705 (Figure 6A). Importantly, when neutralizing anti-IL-6 antibodies were added during the co-culture, this increase in phospho-STAT3 was completely blocked, as was the increase in RAC1B expression (Figure 6A). In contrast, the increase in phospho-STAT3 or RAC1B levels were not affected by the presence of neutralizing antibodies against IL-1β (Figure 6B).

Intriguingly, the above data revealed that neutralizing antibodies against human IL-1β were not effective in blocking the increase in RAC1B when added to the co-cultures (Figure 6B), but that addition of recombinant IL-1β to Caco-2 cells had an effect on RAC1B levels (Figure 5B). Since IL-1β has been described as an inducer of IL-6 expression in colon and breast cancer cells [42,43], we reasoned that IL-1β could trigger IL-6 production by Caco-2 cells. We thus added purified IL-1β, or GM-CSF as a negative control, to the basolateral medium of polarized Caco-2 cells and determined the expression of the IL-6 gene by qRT-PCR. Under these conditions, we observed that IL-1β, but not GM-CSF, triggered a 5-fold increase of IL-6 transcripts after 24 h in Caco-2 cell lysates, whereas RAC1B protein levels only notably increased after 48 h of incubation with IL-1β (Figure 6C,D). These data suggest that the effect of IL-1β on RAC1B levels was indirect, and at least partially mediated through newly synthesized IL-6.

The identification of IL-6 as the main pro-inflammatory factor responsible for inducing the overexpression of tumour-related RAC1B raised the question of whether IL-6 was also involved in the higher RAC1B expression levels that we observed in non-polarized Caco-2 cells in Figure 1C. As shown in Appendix A, the RAC1B protein levels in non-polarized Caco-2 cells did not change after addition of neutralizing anti-IL-6 antibodies, nor did they correlate with increased STAT3 phosphorylation levels. Moreover, RAC1B levels in non-polarized cells did not respond to the addition of purified IL-6 (Appendix A). Thus, the difference between non-polarized and polarized Caco-2 cells observed in Figure 1C does not seem to involve IL-6. 

### 3.4. Interleukin-6 also Stimulates RAC1B Expression in Polarized T84 Colorectal Cells

To demonstrate that the observed effects of co-culture and IL-6 on RAC1B expression can be replicated in another colorectal cell line, we grew a polarized layer of T84 cells as a second model (Figure 7A). First, we determined the effect on RAC1B expression under the three most relevant co-culture conditions identified above: T84 with T84 (control), T84 with CT5.3, and T84 with CT5.3 plus M1. As shown in Figure 7B,C, RAC1B protein levels in polarized T84 also reached a statistically significant increase after 48 h of co-culture with CT5.3 plus M1. This increase was also observed at the respective mRNA level (Figure 7D). The addition of a neutralizing anti-IL-6 antibody to the co-culture condition prevented the observed increase in RAC1B protein levels (Figure 7E,F). Consistently, the addition of purified IL-6 to the basolateral side of polarized T84 cells was sufficient to induce an increase in RAC1B protein and mRNA levels (Figure 7 G–I). In all these experiments, the co-culture or IL-6-mediated effects were reflected by changes in the phosphorylation of STAT3.

## 4. Discussion

Since the seminal work of Rudolf Virchow, pathologists have observed that most solid tumours are infiltrated by inflammatory cells, showing the properties of wounds that fail to heal [6]. Inflammation has been added as a hallmark property of cancer [44], since states of local chronic or pathogen-induced inflammation were found to promote tumour formation [45,46]. In this manuscript, we have studied the effect of pro-inflammatory stimuli from the microenvironment on a model of polarized Caco-2 colorectal cells and found that these respond with the overexpression of tumour-related RAC1B to soluble cytokines secreted by CAFs and M1-like macrophages, particularly to IL-6. 

The Caco-2 cell line derives from an intermediate-stage human colon adenocarcinoma [47] and displays colonocyte and enterocyte characteristics, while the T84 cells were derived from a lung metastasis of a colon carcinoma but retain colonocyte morphology [48]. Both models show the ability to differentiate into an epithelial-like cell layer with basolateral tight junctions. The resulting transepithelial barrier is a characteristic of colon mucosa and better resembles the early events in colon tumorigenesis than conventional monolayer formation on plastic dishes. In this work, both colorectal cell lines were grown on microporous membranes and reached a TEER of up to 600 Ω·cm^2^ after 12–14 days (Figure 1 and Figure 7), a period during which they seal the lateral intercellular spaces through tight junction formation [37,49] and functionally separate the apical and basolateral plasma membranes [50,51,52,53,54]. This is an important characteristic of our study. First, this organization is known to have a profound impact on the wiring of the cells’ signal transduction pathways, because the cell–cell or cell–matrix adhesion complexes assembled by a polarized cell determine the organization of the cytoskeleton and affect the intracellular localization of signalling molecules. In consequence, the signalling response can differ substantially from cells grown as a flat monolayer on a stiff plastic matrix, as previously reviewed by Weaver and colleagues [55]. We confirmed such differences in Figure 1C and Appendix A, where we compared the response of polarized and non-polarized Caco-2 and T84 colorectal cells. Second, the organization of polarized colorectal cells on the microporous membrane allowed for their co-culture with other stromal cell types without being in direct physical contact. Thus, the cells only share soluble factors released into the medium, and a pure colorectal cell extract could be obtained for analysis. 

Using this system, we identified that a combined co-culture with the CAF cell line CT5.3 and M1-type macrophages triggered the most significant increase in RAC1B levels in the polarized Caco-2 colorectal cells. CAFs are fibroblasts activated by the presence of tumour cells and have a significant impact on cancer progression. They can remodel the ECM and secrete growth factors that stimulate cancer cell proliferation and recruit inflammatory cells [56,57,58]. Tumour-associated macrophages (TAMs) include both M1- and M2-like macrophages. In general, higher M1-like infiltrates in an established tumour correlate with a better prognosis, as they can present tumour-cell antigens to effector immune cells. However, at the early stages of tumour development, M1-like macrophages can promote malignant transformation by inducing chronic inflammation [59,60]. In contrast, higher M2-like infiltrates correlate with immune tolerance and poor prognosis, and were primarily found at the invasive front of colorectal tumour sections [61]. 

The growth medium after co-culture with CT5.3 and M1-type macrophages was analysed using a cytokine antibody array and allowed for the identification of three secreted cytokines that were associated with increased RAC1B expression in polarized Caco-2 cells: IL-6, IL-1β, and GM-CSF. All are typical pro-inflammatory cytokines, and frequently have their corresponding receptors expressed in Caco-2 and other colorectal cell lines [62,63,64,65].

A specific increase was detected for GM-CSF in the co-culture condition; however, when this cytokine was added as purified protein to polarized Caco-2 cells, it had no direct effect on the levels of RAC1B (Figure 5C). GM-CSF is a 23 kDa cytokine secreted by macrophages and fibroblasts and part of the inflammatory cascade, by which macrophages become activated. 

The highest change in signal intensity on the cytokine array was seen with IL-6, which increased in co-cultures of Caco-2 with either CAFs or M1-like macrophages alone, but at least 10-fold more in the triple co-culture containing both cell types (see Figure 5A). This mirrored our observation that RAC1B protein levels in Caco-2 cells were slightly increased in the presence of CAFs or M1-like macrophages, but the highest induction was observed in combined co-culture (Figure 3A). Indeed, an IL-6 neutralizing antibody was able to prevent the increase in RAC1B expression that was observed in polarized Caco-2 cells under the triple co-culture condition (Figure 6A). These results were replicated in polarized T84 cells as a second colorectal cell model (Figure 7). Supporting the physiological relevance of our cell system, a role for IL-6 in cancer or inflammation of the colon has been well documented in clinical practice. Increased levels of serum IL-6 have been correlated with poor prognosis in CRC [66] and a variety of cancers, and were associated with CRC tumour size and disease status [67,68,69,70,71]. Anti-IL-6 receptor antibodies that target stromal tissue showed great anti-tumour activity in vivo [72]. Furthermore, IL-6 levels are distinctly elevated in serum and intestinal mucosa of patients with IBD and positively correlated with the severity of inflammation [73]. In colitis-associated colorectal cancer, IL-6 was identified as a critical tumour-promoter during early tumorigenesis by protecting pre-malignant intestinal epithelial cells from apoptosis [74]. IL-6 has been shown to promote the growth of colon-cancer-derived cells in vitro [75]. In polarized Caco-2 cells, IL-6 caused an increase in tight junction permeability by modulating claudin-2 gene expression [76]. 

IL-6 is known to signal through the JAK/STAT3 pathway. Following binding to the IL-6 receptor and gp130 co-receptor, both become tyrosine phosphorylated and recruit the cytosolic transcription factor STAT3 through binding by its SRC homology domain 2 (SH2) domain. Bound STAT3 comes then into proximity with the tyrosine kinase JAK, which phosphorylates STAT3 at Y705, allowing for the nuclear translocation of STAT3 dimers. STAT3 target genes include regulators of cell proliferation and survival, such as cyclin D1, c-MYC and BCL-xL [71]. In polarized Caco-2 and T84 cells, we were able to confirm that they responded to IL-6 with phosphorylation of STAT3. Interestingly, STAT3 was found to be highly phosphorylated in CRC biopsies; however, this was not true for colorectal cell lines including Caco-2, except when these were implanted as xenografts into nude mice, or treated in vitro with IL-6. These studies suggested the requirement of an extracellular stimulus from the microenvironment as a trigger for STAT3 activation [77], in agreement with our observations.

The cytokine array also identified that IL-1β levels in the growth medium were increased, first slightly by the co-culture with CAFs alone, but then over 50-fold in the triple co-culture. IL-1β has been strongly implicated in the acute inflammatory response of immune cells [78], and is a widely used marker gene for M1-like macrophages. Besides its role in acute inflammation, IL-1β is also a key mediator of chronic inflammation-induced pathological changes in IBD [79,80,81,82] and can promote colorectal tumour development [83,84]. 

Unexpectedly however, neutralizing antibodies against IL-1β were poorly efficient in preventing RAC1B overexpression during the co-culture experiments, suggesting an indirect role of IL-1β. Indeed, previous studies found that the pro-tumorigenic role of IL-1β was also related to its ability to induce tumour cells to produce other cytokines, including IL-6 [42,43]. For this, the IL-1β receptor does not activate the JAK/STAT3 pathway but rather stimulates the activation of the p38/JNK and/or the NF-κB pathways [85,86], through a pathway involving the adaptor protein myeloid differentiation primary response gene 88 (MyD88) in association with interleukin 1 receptor-associated kinase (IRAK) [87], leading to activation of TNF receptor-associated factor 6 (TRAF6) and protein kinase TGFβ-activated kinase 1 (TAK1) [88,89]. Actually, we could confirm in our polarized Caco-2 cells that the addition of purified IL-1β stimulated a 5-fold increase in the expression of IL-6 mRNA within 24 h, which could then contribute to the increase in RAC1B protein levels observed after 48 h. Altogether, our findings indicate IL-6 as the key soluble factor that promotes RAC1B overexpression, although we cannot quantify how much IL-6 was contributed by each of the three cell types in the described co-culture condition at present.

The identification of IL-6 raised the question of how Caco-2 and T84 cells relay their response into the observed increase in RAC1B levels. One possibility is that the RAC1B protein was stabilized so that its levels increased. Recent data have provided evidence that RAC1 can be proteolytically degraded following its ubiquitinylation on lysine 147 [90,91,92]; however, RAC1B was found to be a poor substrate for this modification [90]. The ubiquitinylation of RAC1 was proposed to occur in a regulatory loop to counteract excessive activation of RAC1 following the activation of protein kinase JNK by GTP-loaded RAC1, which RAC1B cannot activate. Thus, although we cannot completely exclude that the increase in RAC1B also occurs at the protein level, the current knowledge does not favour this possibility.

An alternative possibility is that the increase in RAC1B levels is a consequence of a shift in alternative splicing towards the alternative RAC1B transcript. Indeed, our data show that the RAC1B mRNA levels also increase in response to the described co-culture conditions (Figure 3 and Figure 7). We have previously shown that, in normal colon mucosa about 95% of the pre-mRNA transcribed from the *RAC1* gene yields the RAC1 mRNA, but in colon tumours, the alternative transcript encoding RAC1B can increase to up to 20% of the total *RAC1* gene-derived mRNA [26,93]. This increase is significant because we found that the resulting RAC1B protein predominantly adopts the active and signalling-competent conformation, in contrast to RAC1, which is maintained, mostly inactive, in cells through interaction with Rho-GDI [26,28,94]. The process of Caco-2 cell polarization somehow appears to repress RAC1B levels and prime these cells to become responsive to pro-inflammatory stimuli, including IL-6. We hypothesize that IL-6-induced signalling communicates in polarized Caco-2 cells with the splicing machinery and alters the balance in alternative splicing of the RAC1 pre-mRNA. For example, the IL-6-induced activation of STAT3 that we observed (see Figure 6 and Figure 7) could change the transcription levels of cellular splicing factors. We previously reported that splicing factors SRSF3 and SRSF1 can inhibit or increase the alternative splicing of RAC1B, respectively [93,95,96]; however, their transcriptional regulation remains to be investigated. If common splicing factors were the target of the IL-6 response, then further alternative splicing events could be affected in the colorectal cells [97,98]. This will require careful genome-wide analyses and will be an interesting avenue for future studies.

## 5. Conclusions

Through the analysis of an epithelial-like layer of polarized Caco-2 cells that was co-cultured in the presence of CAFs and M1-type macrophages, we identified that IL-6 is the main pro-inflammatory factor responsible for inducing the overexpression of tumour-related RAC1B. Since RAC1B can sustain tumour-cell survival or promote escape from oncogene-induced senescence, its induction by external stimuli provides further molecular evidence for the important role that a pro-inflammatory microenvironment can play in tumour progression. Furthermore, the results indicate novel therapeutic opportunities against colorectal cancer development. 

## Figures and Tables

**Figure 1 cancers-14-01393-f001:**
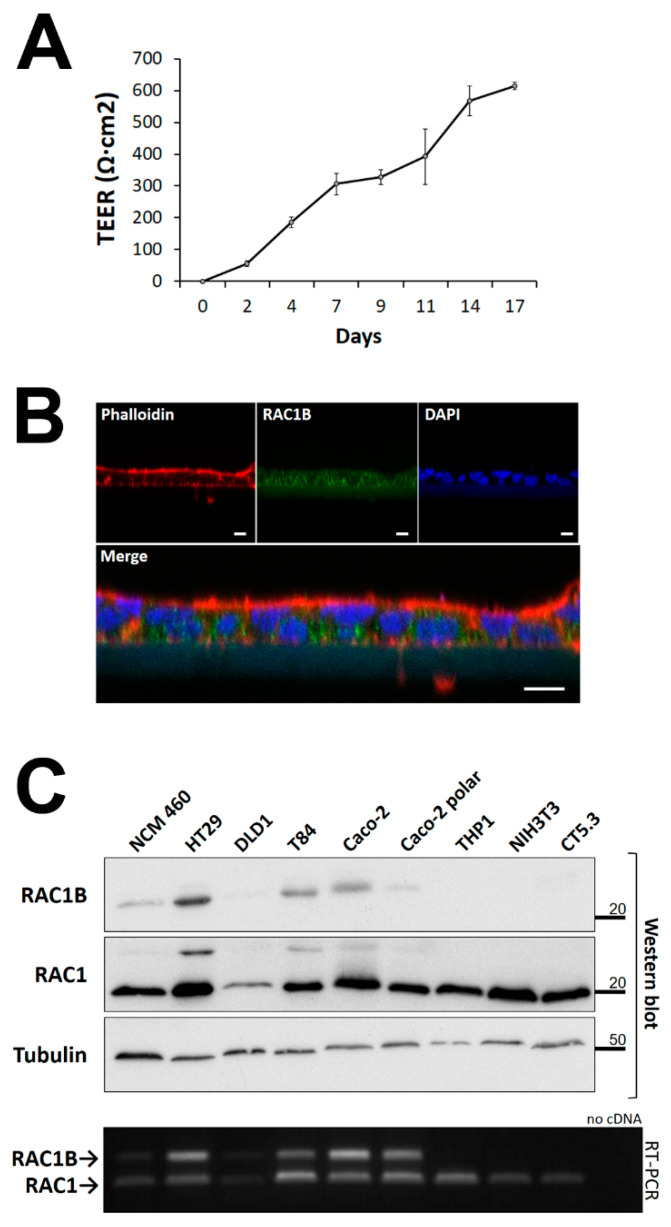
Characterization of RAC1B expression in colorectal cells. (**A**) Caco-2 cells were seeded on filter inserts for the formation of a polarized cell layer and polarization progress was monitored by TEER measurement with a Chopstick Electrode STX2, for up to seventeen days. (**B**) Caco-2 cell polarization and RAC1B localization were analysed by confocal immunofluorescence microscopy at day 14 of culture. Merge is the overlay of three confocal immunofluorescence images, with cell nuclei in blue (DAPI), endogenous RAC1B protein in green, and the actin filament marker phalloidin in red; scale bar = 10 µm. (**C**) Expression of RAC1B and RAC1 protein (Western blot) and mRNA (RT-PCR) in whole cell lysates from colon and stromal cell lines. Proteins were separated by SDS-PAGE and the indicated proteins detected by WB with alpha-Tubulin used as a loading control. Migration position of molecular weight markers is indicated in kDa. Note that the anti-RAC1 antibody also stains RAC1B as a weak band of higher molecular weight. Below, the ethidium bromide-stained agarose gel shows the corresponding amplification of the two *RAC1* gene-derived transcripts of 238 and 181 bp, respectively.

**Figure 2 cancers-14-01393-f002:**
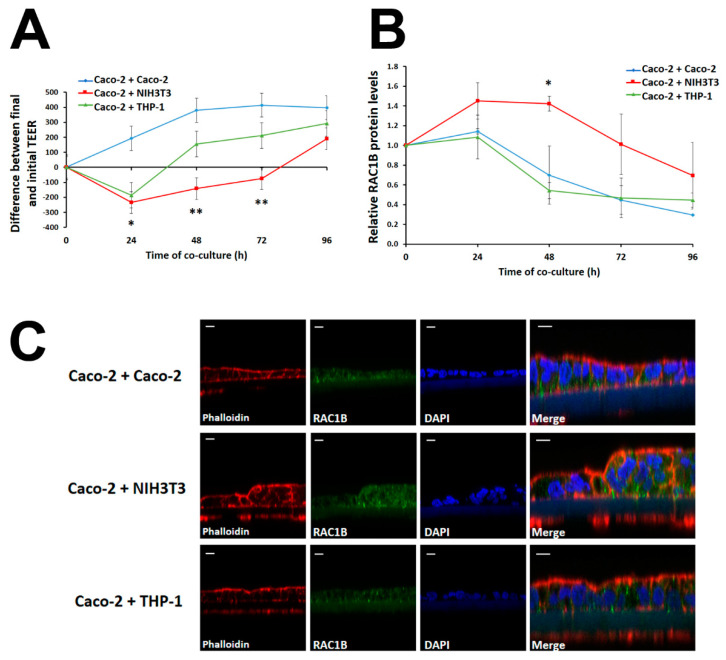
RAC1B levels and changes in polarization of Caco-2 cells under co-culture with stromal cells. Filter inserts with fully polarized Caco-2 monolayers were placed into pre-seeded 24-well plates with Caco-2 (control), NIH3T3 fibroblasts, or THP-1 monocytes. The Caco-2 cells were exposed to different co-culture lengths: 0 h, 24 h, 48 h, 72 h, and 96 h. (**A**) Changes in the polarization state were assessed by the differences between final and initial TEER measurement (ΔTEER) of Caco-2 after different co-culture times. (**B**) RAC1B protein levels in lysates from polarized Caco-2 cells were analysed by SDS-PAGE and WB techniques after different co-cultures times. Data are shown as fold-change relative to the 0 h control and represent means ± SEM, of at least 8 independent experiments. Statistical analysis was carried out with two-way ANOVA tests ((**A**) F = 5.39, *p* = 0.0458; (**B**) F = 4.77, *p* = 0.0576), followed by Bonferroni post hoc tests to assess significant differences from control conditions (Caco-2 + Caco-2) at each timepoint (* *p* < 0.05; ** *p* < 0.01). (**C**) Morphology of polarized Caco-2 cell layers after 48 h of co-culture with stromal cells. Cell polarization and RAC1B expression and localization were analysed by confocal immunofluorescence microscopy. Merge is the overlay of three confocal immunofluorescence images, which detected cell nuclei in blue (DAPI), the localization of endogenous RAC1B protein in green, and actin (phalloidin) in red; scale bars = 10 µm.

**Figure 3 cancers-14-01393-f003:**
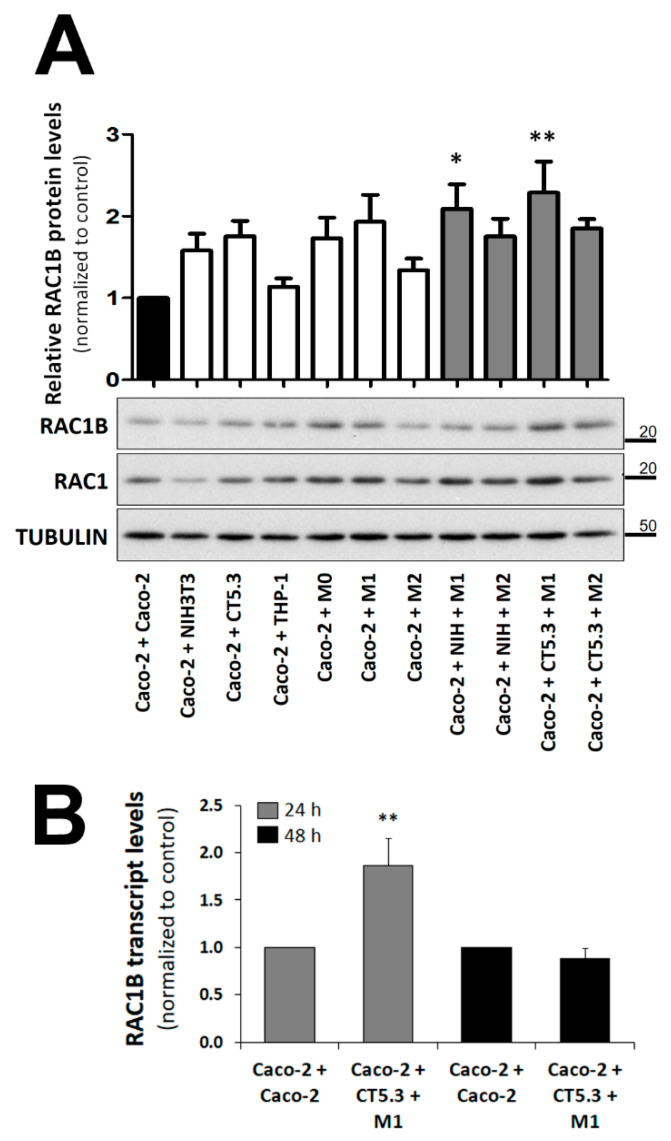
Effect of different co-culture conditions on the RAC1B expression levels in polarized Caco-2 cells. Polarized Caco-2 cells were co-cultured with the indicated control or stromal cells and lysed after 48 h of culture. (**A**) Proteins from the whole-cell lysates were analysed by SDS-PAGE and the indicated proteins detected by WB. The control of the assay corresponds to the co-culture of Caco-2 with Caco-2, and the α-tubulin protein served as a loading control. Migration position of molecular weight markers is indicated in kDa. The corresponding quantification of RAC1B levels in Caco-2 was obtained from at least three independent biological replicate experiments. Band intensities were measured and normalized to tubulin levels. Then, the RAC1B/RAC1 ratio was calculated (as both are derived from the same pre-mRNA transcript). Data are shown as the fold-change in the RAC1B/RAC1 ratio relative to control and represent means ± SEM, of at least 5 independent experiments. Statistical analysis was carried out with a one-way ANOVA test (F = 2.82; *p* < 0.0056), followed by a Tukey’s post hoc test. * or ** significantly different from the corresponding control (Caco-2 + Caco-2) with *p* < 0.05 or *p* < 0.01, respectively. (**B**) Total RNA was extracted from polarized Caco-2 cells after 24 h and 48 h of the indicated co-culture, cDNA synthetized, and RAC1 and RAC1B transcripts amplified by qRT-PCR. Data are shown as fold-change relative to control of the RAC1B/RAC1 ratio (both derive from the same pre-mRNA) and represent mean values ± standard error of the mean (SEM), of 5 independent experiments. Statistical analysis was carried out with a one-way ANOVA test (F = 79.69, *p* < 0.001), followed by Tukey’s post hoc test; ** significantly different from the corresponding control with *p* < 0.01.

**Figure 4 cancers-14-01393-f004:**
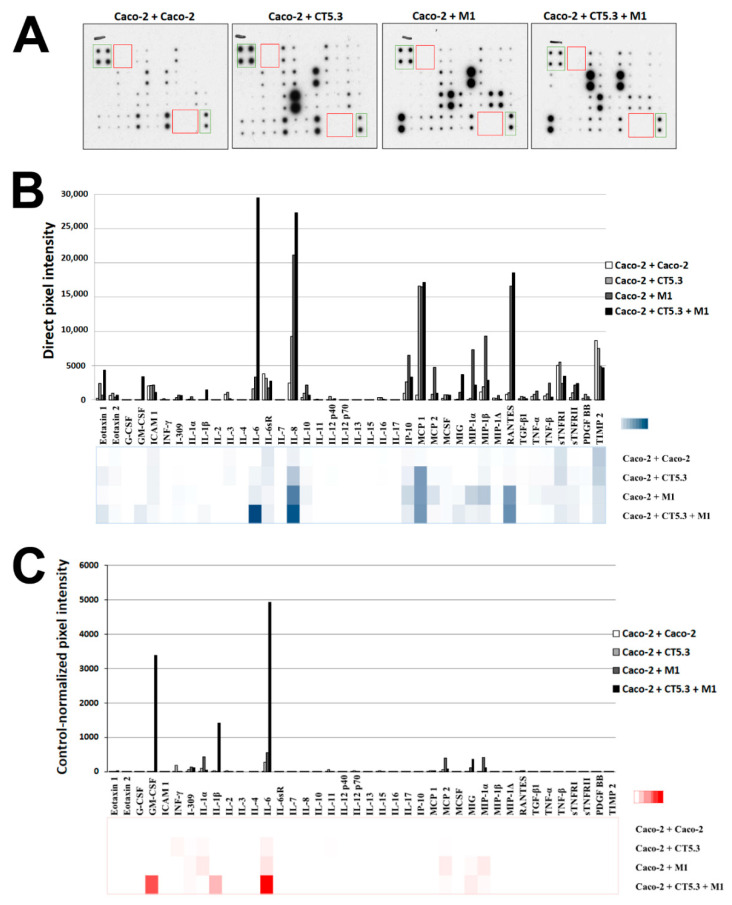
Identification of cytokines from the co-culture medium. Following the co-culture of polarized Caco-2 cells with the indicated cell types, the corresponding media were incubated with antibody arrays and their cytokine composition detected by a WB-like procedure. (**A**) Shown are the observed signal intensities, with positive and negative control spots marked by green or red boxes, respectively. (**B**) Graphic display of the direct pixel intensities observed for each cytokine under the four conditions. (**C**) Graphic display of the pixel intensities after normalization to the background values obtained in the control co-culture with Caco-2 itself.

**Figure 5 cancers-14-01393-f005:**
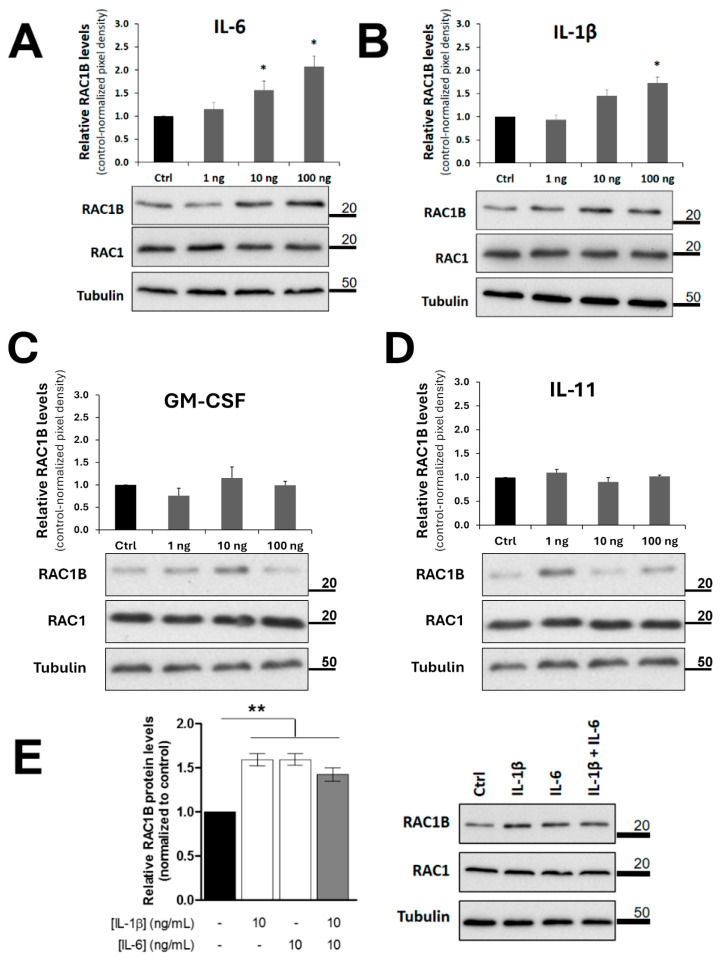
RAC1B protein levels in polarized Caco-2 cells after addition of purified candidate cytokines. Polarized Caco-2 cells were grown on filter inserts and then the indicated (**A**–**D**) purified cytokines or (**E**) cytokine combinations were added to the basolateral growth medium. Cells were lysed after 48 h, proteins analysed by SDS-PAGE and the indicated proteins detected by WB. Detection of the α-tubulin protein served as a loading control. Data are shown as fold change in RAC1B protein levels relative to control (addition of antibody solvent PBS to the medium) and represent mean values ± SEM, of at least 3 independent experiments. Statistical analysis was carried out with a one-way ANOVA test (F = 39.04, *p* < 0.001), followed by Tukey’s post hoc tests; * or ** significantly different from the corresponding control (PBS) with *p* < 0.05 or *p* < 0.01, respectively.

**Figure 6 cancers-14-01393-f006:**
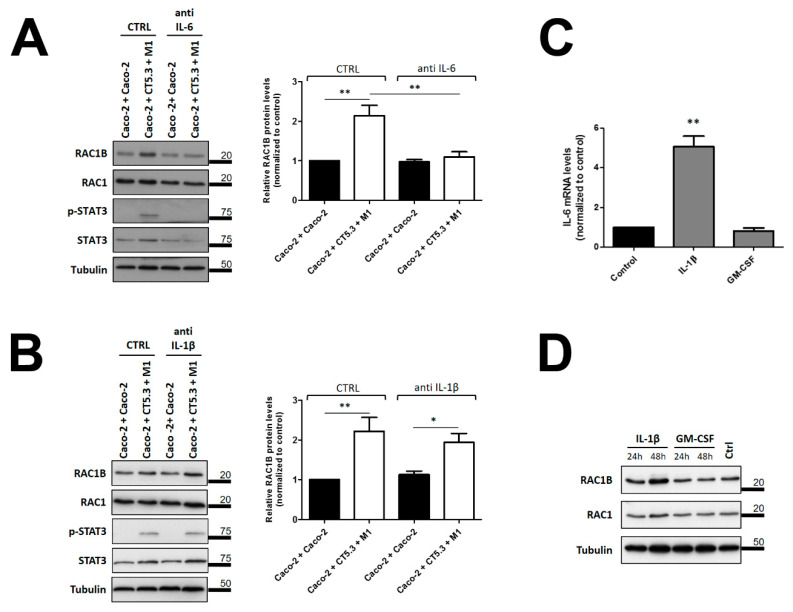
Effect of neutralizing antibodies against IL-6 or IL-1β on phospho-STAT3 and RAC1B protein levels in Caco-2 cells. Polarized Caco-2 cells were co-cultured with the indicated control or with CT5.3 and M1 macrophages, in the presence of 500 ng/mL of neutralizing antibody against (**A**) human-IL-6 or (**B**) human IL-1β. Cells were lysed after 48 h of culture, proteins from the whole-cell lysates analysed by SDS-PAGE and the indicated proteins detected by WB. Detection of the tubulin protein served as a loading control and migration position of molecular weight markers is indicated in kDa. The control of the assay corresponds to the co-culture of Caco-2 with Caco-2 cells in the presence of antibody solvent (PBS). Graphics show the fold change in RAC1B protein relative to the control co-culture and represent mean values ± SEM of at least 3 independent experiments. Statistical analysis was carried out with a one-way ANOVA test ((**A**): F = 13.21, *p* < 0.001; (**B**): F = 9.15, *p* < 0.001), followed by Tukey’s post hoc tests; * or ** significantly different from the corresponding control condition (Caco-2 + Caco-2 or PBS) with *p* < 0.05 or *p* < 0.01, respectively. (**C**,**D**) Effect of purified IL-1β on (**C**) endogenous IL-6 mRNA expression and (**D**) RAC1B protein in Caco-2 cells. Polarized Caco-2 cells were incubated in the presence of 10 ng/mL of indicated purified cytokines. The control of the assay corresponds to Caco-2 cells incubated with antibody solvent (PBS). In (**C**), the endogenous IL-6 transcript levels in Caco-2 cells are shown after 24 h, as determined by qRT-PCR. Data are shown as fold change relative to control and represent mean values ± SEM, of at least 3 independent experiments. Statistical analysis was carried out with a one-way ANOVA test [F = 79.69, *p* < 0.001], followed by Tukey’s post hoc tests; ** significantly different from the corresponding control with *p* < 0.01. In (**D**), the corresponding IL-1 β effect on RAC1B and control protein levels after 24 and 48 h are shown in whole-cell lysates by WB.

**Figure 7 cancers-14-01393-f007:**
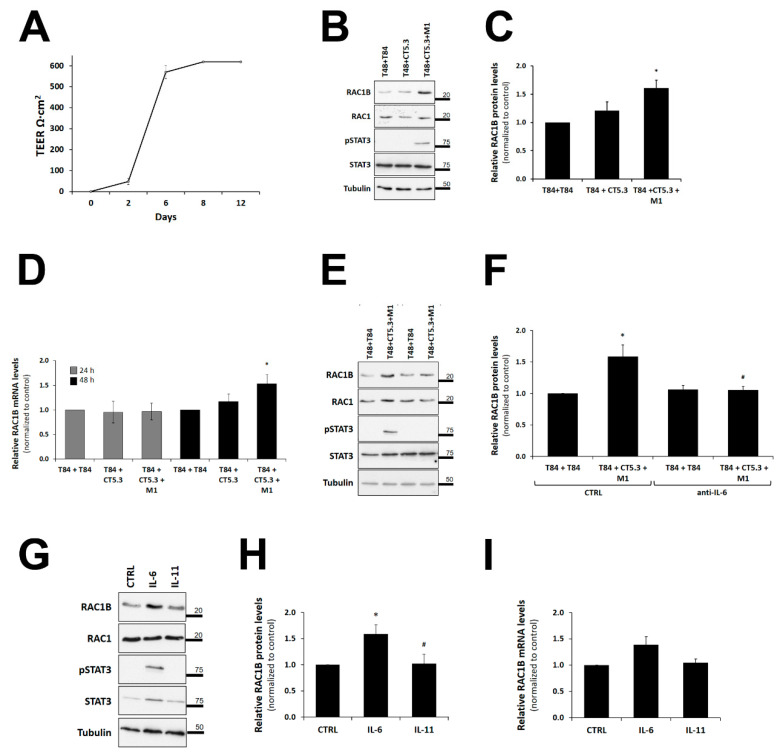
Effect of pro-inflammatory co-culture conditions or of purified IL-6 on RAC1B levels in polarized T84 cells. (**A**) Progress of cell polarization of T84 cells grown on filter inserts was monitored by TEER measurement with a Chopstick Electrode STX2, during up to twelve days. (**B**–**D**) Effect of co-culture conditions on RAC1B levels. Polarized T84 cells were co-cultured with control T84 or the indicated stromal cells and lysed after 24 h or 48 h of culture. (**B**) Proteins from the whole-cell lysates were analysed by SDS-PAGE and the indicated proteins detected by WB. Migration position of molecular weight markers is indicated in kDa. The control of the assay corresponds to the co-culture of T84 with T84 cells. Corresponding quantification of (**C**) RAC1B protein and (**D**) transcript levels by qRT-PCR in T84 cells obtained from at least six independent biological replicate experiments. Band intensities were measured and the RAC1B/RAC1 ratios calculated as before. Data are shown as fold-change of the RAC1B/RAC1 ratio relative to control and represent mean values ± SEM of 5 independent experiments. Statistical analysis was carried out with a one-way ANOVA test (F = 5.49, *p* = 0.0113 for (**C**) and F = 4.82, *p* = 0.0219 for (**D**)) followed by Tukey’s post hoc tests; * significantly different from the corresponding control (T84 + T84) with *p* < 0.01 in (**C**) or *p* < 0.05 in (**D**). (**E**,**F**) Effect of neutralizing antibodies against IL-6 on RAC1B and phospho-STAT3 protein levels. Polarized T84 cells were co-cultured with the indicated control or with CT5.3 and M1 macrophages, in the presence of 500 ng/mL of neutralizing antibody against human-IL-6. Graphic shows the fold change in RAC1B protein relative to control and represent mean values ± SEM of at least 5 independent experiments. Statistical analysis was carried out with a one-way ANOVA test (F = 9.62, *p* = 0.0003), followed by Tukey’s post hoc tests; * or # significantly different with *p* < 0.01 from the corresponding controls (T84 + T84) or (T48 + CT5.3 + M1), respectively. (**G**–**I**) RAC1B levels after addition of purified cytokines. Polarized T84 cells were grown on filter inserts and then 10 ng/mL of purified IL-6 or IL-11 (negative control) were added to the basolateral growth medium, and cells lysed after 48 h. (**G**) Proteins were analysed by SDS-PAGE and the indicated proteins detected by WB. Graphs show data as fold change in (**H**) RAC1B protein or (**I**) transcript levels by qRT-PCR relative to control (addition of antibody solvent PBS to the medium) and represent mean values ± SEM, of at least 6 independent experiments. Statistical analysis was carried out with a one-way ANOVA test (F = 4.81, *p* = 0.0112), followed by Tukey’s post hoc tests; * and # indicate significant differences (*p* < 0.05) from the control (PBS) and IL-6, respectively.

## Data Availability

This study did not report any data not shown in the manuscript.

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
