# Peer review of "Pro-Inflammatory Cytokines Trigger the Overexpression of Tumour-Related Splice Variant RAC1B in Polarized Colorectal Cells"

_cancers, 2022, doi:10.3390/cancers14061393_

Round 1

Reviewer 1 Report

This work by Pereira et al studies the increase in expression of Rac1b in response to co cultures with fibroblasts and M1 macrophages. They then establish that this is in response to IL6. The study design is logical and the authors have data proposing a mechanism behind this increase in Rac1b expression. I have no major concerns and believe that this will be of interest to readers. I do have a few minor concerns and comments listed below. But in general this is a nicely written and well studied paper.

Minor concerns:

1) The use of the Rac1b antibody for IF needs to be validated . THerefore the authors should include their validation of the antibody (knockdown or knockout) or a reference included of a paper that validates the antibody for this technique. My main concern is becuase this antibody is a polyclonal and may therefore also bind other targets. In my experience it produces non-specific bands on Western blot which indicates it may have other targets for IF.

2) Representative Western blots of Figure 2B should be included in supplemental.

3) The bar charts of Western blot densitometry would benefit from overlaying of the individual points.

4) STAT3 is spelt incorrectly in the discussion, 7th paragraph ("IL-6 is known to signal through the JAK/STA3 pathway")

Author Response

This work by Pereira et al studies the increase in expression of Rac1b in response to co cultures with fibroblasts and M1 macrophages. They then establish that this is in response to IL6. The study design is logical and the authors have data proposing a mechanism behind this increase in Rac1b expression. I have no major concerns and believe that this will be of interest to readers. I do have a few minor concerns and comments listed below. But in general this is a nicely written and well studied paper.

 Minor concerns:

1) The use of the Rac1b antibody for IF needs to be validated . THerefore the authors should include their validation of the antibody (knockdown or knockout) or a reference included of a paper that validates the antibody for this technique. My main concern is becuase this antibody is a polyclonal and may therefore also bind other targets. In my experience it produces non-specific bands on Western blot which indicates it may have other targets for IF.

The authors acknowledge the reviewer’s concern and have consulted several papers in which this antibody was used for immunohistochemistry studies in different tissue types. Thus, as suggested by the reviewer, we included these references (doi: 10.1091/mbc.E12-02-0166 and 10.1158/1541-7786.MCR-13-0557-T, now refs 35 and 36 of the manuscript) in the immunofluorescence subsection of the Material and methods.

2) Representative Western blots of Figure 2B should be included in supplemental.

In agreement with the reviewer’s suggestion, we have added a new Figure S2 into Supplementary file S2, which shows a representative Western blot for the quantification shown in Fig 2.

3) The bar charts of Western blot densitometry would benefit from overlaying of the individual points.

We tested the suggested overlapping display in one of the charts and found the results more difficult to interpret. Moreover, the remaking of all plots to comport this layout would be difficult in the allotted time-frame for revision. Since many of the articles published in Cancers follow the same bar chart layout, we believe their current format is also suitable.

4) STAT3 is spelt incorrectly in the discussion, 7th paragraph ("IL-6 is known to signal through the JAK/STA3 pathway")

The spelling error has been corrected

Reviewer 2 Report

In this manuscript, Pereira and colleagues investigated the expression of a RAC1 alternative splicing variant, RAC1B, in polarized colorectal cancer cells (Caco-2) after co-culture with different stromal cells. In this model, the authors first analyzed RAC1B expression by RT-PCR, Western blot and confocal fluorescence microscopy after cultivating tumor cells on a microporous filter membrane. Next, the authors demonstrated that, upon co-culturing for 48 hours, RAC1B protein levels increased in the presence of a fibroblast cell line but decreased under control conditions. The strongest effect on increased RAC1B expression in Caco-2 cells was observed after co-culture with patient-derived CAFs and with human M1-like macrophages. To analyze which soluble factors mediate the increased RAC1B levels, the authors investigated cell culture supernatants using 40-cytokine-antibody arrays, and identified IL-6, IL-1β and GM-CSF. Further validation revealed IL-6 as the key factor stimulating RAC1B expression, which also leads to STAT3 phosphorylation in Caco-2 cells upon co-culture with CAFs and M1 macrophages. All these results have been subsequently replicated in a second colorectal cancer cell line, T84.

This study is well done and the results are clearly described. In addition, the authors conducted reasonable control experiments with neutralizing antibodies leading to a reversed effect in RAC1B expression, underlining the robustness of their data. I have no concerns about the experiments or no major suggestions to improve the findings. The main weakness of this article is the discussion (see major points of critique below).

Major points of critique

  • Discussion in general: The vast majority of the references/citations in this section are very old. This clearly needs to be addressed!
  • Discussion, page 15, third section: References for CAFs are missing, some key papers have to be cited here.
  • Results 3.1, second paragraph/Figure 1C: The authors showed RAC1B expression in a panel of four colon cancer cell lines. For the subsequent experiments (Figures 2 – 6) they used Caco-2. It should be clearly explained why the authors chose this cell line.

Minor points of critique:

  • Results 3.1, end of second paragraph: the author should include a reference for their statement “… consistent with previous report that related its overexpression to tumour progression.”
  • Discussion, page 16, end of first section: Reference is not displayed in Cancers-Style „IL-6 has been shown to promote the growth of colon cancer-derived cells in vitro (Schneider et al., 2000)“.
  • The Western blot images in the main figures show no size markers.
  • Figure S1: The marker lines are not labeled.
  • Add protein size to all Western blot images.
  • Figure S2: Label the length of the size bar.

Author Response

 Major points of critique

  • Discussion in general: The vast majority of the references/citations in this section are very old. This clearly needs to be addressed!

Concerning the citation of well-established concepts in the Discussion, such as the role of inflammation in cancer or the characterization of Caco-2 cell growth as polarized cell layer, we feel it appropriate to give credit to original and seminal papers on these issues. These happen to be produced some time ago. Thus, we maintained these ‘old´ references, but have added some more recent examples, including refs 45 and 46 on the link between inflammation and cancer, or 53 and 54 on the use of polarized cell models.  Regarding the expression of pro-inflammatory cytokine receptors in Caco-2 cells we also added two more recent citations (63 and 64).

  • Discussion, page 15, third section: References for CAFs are missing, some key papers have to be cited here.

We assume, the reviewer refers to page 25, not 15, of the manuscript file, where CAFs are discussed. We agree with the suggestion and have added three recent key papers (ref 56-58).

  • Results 3.1, second paragraph/Figure 1C: The authors showed RAC1B expression in a panel of four colon cancer cell lines. For the subsequent experiments (Figures 2 – 6) they used Caco-2. It should be clearly explained why the authors chose this cell line.

In order to better explain our choice, we have modified the first two paragraphs and made the following improvements:

We tested the above mentioned hypothesis that microenvironmental stimuli could be the trigger for changes in the expression of RAC1B in colorectal tumour cells. For this, a cell line able to form a polarized epithelial-like cell layer was used and Caco-2 adenocarcinoma cells are the most widely used model to represent the organization and signaling in the epithelial barrier of the mucosa [35,36]. As shown in Figure 1A-B, Caco-2 cells grown on a microporous filter membranes formed a fully polarized cell layer with high transepithelial electrical resistance (TEER) and an apical actin belt. RAC1B was detected at the basolateral membranes. The layer contained ZO-1-positive tight junctions and increased E-cadherin expression (Supplementary Figure S1). The expression of endogenous RAC1B protein was detected by Western blot (WB) in whole cell lysates from polarized Caco-2 cells, as well as in non-polarized Caco-2. Although RAC1B can be readily detected in other colon cell lines (HT29, T84 or NCM460), it was virtually absent in stroma-derived cell types, including fibroblasts (NIH3T3, CT5.3) and monocytes (THP1) (Figure 1C). Interestingly, we observed that RAC1B levels were lowest in normal NCM460 colonocytes [37] and in polarized Caco-2 cells, consistent with previous reports that related its overexpression to tumour progression [26,31,40,41].

Minor points of critique:

  • Results 3.1, end of second paragraph: the author should include a reference for their statement “… consistent with previous report that related its overexpression to tumour progression.”

We have added the respective references [26,31,40,41]

  • Discussion, page 16, end of first section: Reference is not displayed in Cancers-Style „IL-6 has been shown to promote the growth of colon cancer-derived cells in vitro (Schneider et al., 2000)“.

This reference has now been correctly cited [75]

  • The Western blot images in the main figures show no size markers.

All Western blot images were revised and now indicate the position of molecular weight markers.

  • Figure S1: The marker lines are not labeled. Add protein size to all Western blot images.

All Western blot images were revised and now indicate the position of molecular weight markers.

  • Figure S2: Label the length of the size bar.

The magnification of these images has now been included into the figure legend. In addition, the size bars visible in all confocal images were set to 10 µM and referred to in the respective figure legends.

Reviewer 3 Report

Dear Authors,

Major issue:

  1. A very similar work has been presented at the conferences by the author not mentioned in the paper, thus it triggers an ethical concern for me (source: https://orcid.org/0000-0003-1934-4279). Please, explain it.
  2. Provide a full names for the first time, even in abstract e.g. RAC1, BRAF, RTqPCR etc., similarly in the introduction section and other parts of manuscript.
  3. Figure 1- provide the specific name of tubulin used in the figure
  4. In the figure legends declare what the data are presented e.g. mean ±SD or so? of how many replicates?
  5. The authors declare that figure 1 does presents the results form RTqPCR however they did not calculate the expression of gene but only provide the amount of cDNA on gels? Thus it is not a RTqPCR but only amplification.
  6. Figure 3B. The authors declare 5 replicates, no SD visible for some samples, how could it be explained?
  7. The replicates of Western lot were not provided in the supplementary file (at least WB should be done twice to do statistics)

Author Response

Major issue:

  1. A very similar work has been presented at the conferences by the author not mentioned in the paper, thus it triggers an ethical concern for me (source: https://orcid.org/0000-0003-1934-4279). Please, explain it.
    The work referred to is a poster presentation with preliminary data in the personal ORCID record of one of the co-authors. The authors do not see how this can constitute any ethical concern.

  2. Provide a full names for the first time, even in abstract e.g. RAC1, BRAF, RTqPCR etc., similarly in the introduction section and other parts of manuscript.
    Full names of KRAS, BRAF, RAC1, NF-kB and RT-PCR have now been added the first  time they appear, as requested, except for the abstract due to the imposed word limit.

  3. Figure 1- provide the specific name of tubulin used in the figure
    The legend to Fig.1 already states that alpha-tubulin was used.

  4. In the figure legends declare what the data are presented e.g. mean ±SD or so? of how many replicates?
    All Figure legends now contain a sentence such as: ‘Data are shown as fold-change relative to control of the RAC1B/RAC1 ratio and represent mean values ± standard error of the mean (SEM) of 5 independent experiments’.

  5. The authors declare that figure 1 does presents the results form RTqPCR however they did not calculate the expression of gene but only provide the amount of cDNA on gels? Thus it is not a RTqPCR but only amplification.
    The qualitative RT-PCR image was added to Fig. 1C in response to a request from previous Reviewer 2. Accordingly, the corresponding legend states: ‘(C) Expression of RAC1B and RAC1 protein (Western blot) and mRNA (RT-PCR) in whole cell lysates from colon and stromal cell lines.’ Thus, no RT-qPCR data are presented in this figure. Figures showing RT-qPCR are 3B, 6C, 7D and 7I and this is indicated in their respective legends.
  6. Figure 3B. The authors declare 5 replicates, no SD visible for some samples, how could it be explained?
    In Figure 3B, data are expressed relative to the controls at 24 h or 48 h, which are set as value ‘1’, and therefore have no SEM value associated.

  7. The replicates of Western blot were not provided in the supplementary file (at least WB should be done twice to do statistics)
    As stated in each figure legend, all Western blots were repeated three or more times (as indicated in the corresponding figure legends), and the corresponding mean values and associated statistics are presented in graphic form. The supplementary file requested by the journal contains the original images for the Western blot replicate that best represented the mean values and which were chosen to assemble the manuscript figures.

Round 2

Reviewer 3 Report

  1. A very similar work has been presented at the conferences by the author not mentioned in the paper, thus it triggers an ethical concern for me (source: https://orcid.org/0000-0003-1934-4279). Please, explain it.
    The work referred to is a poster presentation with preliminary data in the personal ORCID record of one of the co-authors. The authors do not see how this can constitute any ethical concern.

Dear Authors,

I still see a concern in this issue. First of all the author of the poster is not mention in the publication. The publication includes four co-authors, no one of these authors is included in the poster information on ORCID. Thus, I see an ethic concern in this issue and it should be explained before publication of your manuscript.

  1. Provide a full names for the first time, even in abstract e.g. RAC1, BRAF, RTqPCR etc., similarly in the introduction section and other parts of manuscript.
    Full names of KRAS, BRAF, RAC1, NF-kB and RT-PCR have now been added the first time they appear, as requested, except for the abstract due to the imposed word limit.

The instruction to Authors stays “Acronyms/Abbreviations/Initialisms should be defined the first time they appear in each of three sections: the abstract; the main text; the first figure or table”. Thus, I ask you once again to provide a full names for the first time in the abstract.

  1. Figure 1- provide the specific name of tubulin used in the figure
    The legend to Fig.1 already states that alpha-tubulin was used.

Still, I ask you to provide this information in the Figure.

  1. In the figure legends declare what the data are presented e.g. mean ±SD or so? of how many replicates?
    All Figure legends now contain a sentence such as: ‘Data are shown as fold-change relative to control of the RAC1B/RAC1 ratio and represent mean values ± standard error of the mean (SEM) of 5 independent experiments’.

Thank you.

  1. The authors declare that figure 1 does presents the results form RTqPCR however they did not calculate the expression of gene but only provide the amount of cDNA on gels? Thus it is not a RTqPCR but only amplification.
    The qualitative RT-PCR image was added to Fig. 1C in response to a request from previous Reviewer 2. Accordingly, the corresponding legend states: ‘(C) Expression of RAC1B and RAC1 protein (Western blot) and mRNA (RT-PCR) in whole cell lysates from colon and stromal cell lines.’ Thus, no RT-qPCR data are presented in this figure. Figures showing RT-qPCR are 3B, 6C, 7D and 7I and this is indicated in their respective legends.

In the figure legend you mentioned :”(B) Total RNA was extracted from polarized Caco-2 cells after 24 h and 48 h of the indicated co-culture, cDNA synthetized, and RAC1 and RAC1B transcripts amplified by qRT-PCR” thus I was asking about the quantitative method. I do not see a point in providing such an information here, if you quantify the expression by RT-PCR.

  1. Figure 3B. The authors declare 5 replicates, no SD visible for some samples, how could it be explained?
    In Figure 3B, data are expressed relative to the controls at 24 h or 48 h, which are set as value ‘1’, and therefore have no SEM value associated.

OK

  1. The replicates of Western blot were not provided in the supplementary file (at least WB should be done twice to do statistics)
    As stated in each figure legend, all Western blots were repeated three or more times (as indicated in the corresponding figure legends), and the corresponding mean values and associated statistics are presented in graphic form. The supplementary file requested by the journal contains the original images for the Western blot replicate that best represented the mean values and which were chosen to assemble the manuscript figures.

The journal requirements says: “A single PDF file or a zip folder including all the original images reported in the main figure and supplemental figures should be prepared. Authors should annotate each original image, corresponding to the figure in the main article or supplementary materials, and label each lane or loading order. All experimental samples and controls used for one comparative analysis should be run on the same blot/gel image. For quantitative analyses, please provide the blots/gels for each independent biological replicate used in the analysis.” Thus, I request yu to provide all western blots used for analysis.

Author Response

A very similar work has been presented at the conferences by the author not mentioned in the paper, thus it triggers an ethical concern for me (source: https://orcid.org/0000-0003-1934-4279).

I still see a concern in this issue. First of all the author of the poster is not mention in the publication. The publication includes four co-authors, no one of these authors is included in the poster information on ORCID. Thus, I see an ethic concern in this issue and it should be explained before publication of your manuscript.

We now understood better the reviewer’s concern. We attach for the reviewer’s inspection the corresponding meeting abstract book (see poster P4, page 70). As can be seen, the authors from this manuscript are indeed the authors of this poster, except that the data provided by our colleague Vânia Gonçalves were not included into the final manuscript version. Thus, her name is not listed as a co-author of the submitted manuscript but she was co-author of the poster.

 The instruction to Authors stays “Acronyms/Abbreviations/Initialisms should be defined the first time they appear in each of three sections: the abstract; the main text; the first figure or table”. Thus, I ask you once again to provide a full names for the first time in the abstract.

We have now included the abbreviations also into the abstract and the legend to Figure 1, as marked by track changes in the revised manuscript.

Figure 1- provide the specific name of tubulin used in the figure
 Still, I ask you to provide this information in the Figure.

We have now included this information into the Figure and replaced Figure 1 with the new version

In the figure legend you mentioned :”(B) Total RNA was extracted from polarized Caco-2 cells after 24 h and 48 h of the indicated co-culture, cDNA synthetized, and RAC1 and RAC1B transcripts amplified by qRT-PCR” thus I was asking about the quantitative method. I do not see a point in providing such an information here, if you quantify the expression by RT-PCR.

The respective figure legend sentence has been simplified into: Expression of RAC1B and RAC1 protein (Western blot) and mRNA (RT-PCR) in whole cell lysates from colon and stromal cell lines. We thus feel that the reviewer’s concern no longer applies.

 The journal requirements says: “A single PDF file or a zip folder including all the original images reported in the main figure and supplemental figures should be prepared. Authors should annotate each original image, corresponding to the figure in the main article or supplementary materials, and label each lane or loading order. All experimental samples and controls used for one comparative analysis should be run on the same blot/gel image. For quantitative analyses, please provide the blots/gels for each independent biological replicate used in the analysis.” Thus, I request you to provide all western blots used for analysis.

All original Western Blot images have been provided, and correspond to each Western Blot figure shown in the main article or supplementary materials.

This manuscript is a resubmission of an earlier submission. The following is a list of the peer review reports and author responses from that submission.

Round 1

Reviewer 1 Report

The authors present a well written manuscript on the overexpression of RAC1B induced by cytoikines. The authors performed various experiments that are well described and performed. The results support the hypothesis. 

These data are novel in the quickly evolving research on tumor microenvironment and influence of cytokines on tumor microenvironment and thus on tumor growth.

Inflammation is known to induce tumor growth and these results offer further proof for this notion. Identification of specific molecules such as RAC1B in the pathway can potentially lead to novel treatment possibilities.

Thus, in my opinion, this manuscript merits publication in Cancers.

Author Response

We thank the reviewer #1 for this positive evaluation.

Reviewer 2 Report

This manuscript studied the role of RAC1B during cancer progression. The authors found that RAC1B is upregulated during epithelial-to-cancer differentiation and by co-cultured medium from fibroblast and macrophages. In addition, IL-6 secreted from macrophages upregulates RAC1B through Stat3 activation. This manuscript shows some interesting observations, but some specific concerns are listed below.

  1. Based on previous paper (J Biol Chem. 2014 Oct 3; 289(40): 27386–27399. doi: 10.1074/jbc.M114.589432), expression of ESRP1 and ESRP2 should be examined.
  2. Anti-IL-6 antibody suppressed Stat3 activation and RAC1B induction after cocultivation with macrophages (Fig. 6A), but it is unclear why RAC1B expression become higher in non-polarized Caco2 cells than polarized cells (Fig. 1C). Is IL-6 also involved in this process? The authors should show phosphorylation status of Stat3 in figure 1C and the effect of anti- IL-6 antibody and Stat3 siRNA in this experiment.
  3. Although RAC1B is an alternative splicing variant of RAC1, the authors did not show this splicing event. Do IL6 and epithelial-to-cancer differentiation alter splicing variants? To determine this, mRNA levels of RAC1B and RAC1 should be also shown by conventional PCR analysis in non-polarized and polarized Caco2 cells (Fig 1C), and Caco2 cells upon cocultured with macrophages and IL6 (Fig. 5 and 6). Only protein levels are not sufficient to evaluate the splicing events.

Author Response

  1. Based on previous paper (J Biol Chem. 2014 Oct 3; 289(40): 27386–27399. doi: 10.1074/jbc.M114.589432), expression of ESRP1 and ESRP2 should be examined.

We thank the reviewer for this attentive comment. Indeed, we are eager to study these but also other splicing factors such as SRSF1 and SRSF3, regarding both their expression and phosphorylation levels following IL-6 stimulation of Caco-2 and T84 cells. However, we feel these experiments are out of scope for this manuscript, in which we identify that signals produced by a pro-inflammatory microenvironment are able to stimulate RAC1B expression in polarized colorectal cells. How these signals talk to the molecular splicing machinery will have to be further explored in the future in a more detailed and separate study.

  1. Anti-IL-6 antibody suppressed Stat3 activation and RAC1B induction after cocultivation with macrophages (Fig. 6A), but it is unclear why RAC1B expression become higher in non-polarized Caco2 cells than polarized cells (Fig. 1C). Is IL-6 also involved in this process? The authors should show phosphorylation status of Stat3 in figure 1C and the effect of anti- IL-6 antibody and Stat3 siRNA in this experiment.

This is a very interesting suggestion that we pursued in new experiments. We have compared between polarized and non-polarized Caco-2 cells their response to IL-6 stimulation and to IL-6 blocking. We found that the high expression of RAC1B in non-polarized monolayers was not correlated with an increased level of phospho-STAT3 in these cells, although comparable levels of total STAT3 were expressed under both growth conditions. Moreover, neither stimulation of non-polarized cells with recombinant IL-6 nor blocking with anti-IL-6 antibody affected their RAC1B levels. Thus, an increase in IL-6 production or signalling by non-polarized cells cannot account for the observed higher RAC1B levels, when compared to polarized cells.

This suggests that the process of Caco-2 cell polarization somehow represses RAC1B levels and primes these cells to become responsive to pro-inflammatory stimuli, including IL-6.

In the revised manuscript, these data cannot be included directly into Figure 1C, as this would alter the logic flow of arguments presented in the manuscript. We thus describe these results after the paragraph on the identification of IL-6 as a driver of the observed changes in RAC1B expression, at the end of section 3.3, and included the data as Supplementary Figure 3.

In particular, the newly added text on page 20 says:

The identification of IL-6 as the main pro-inflammatory factor responsible for inducing overexpression of tumour-related RAC1B raised the question whether IL-6 was also involved in the higher RAC1B expression levels that we observed in non-polarized Caco-2 cells in Figure 1C. As shown in Figure S3A, the RAC1B protein levels in non-polarized Caco-2 cells did not change after addition of neutralizing anti-IL-6 antibodies, nor did they correlate with increased STAT3 phosphorylation levels. Moreover, RAC1B levels in non-polarized cells did not respond to the addition of purified IL-6 (Figure S3B). Thus, the difference between non-polarized and polarized Caco-2 cells observed in Figure 1C does not seem to involve IL-6.

  1. Although RAC1B is an alternative splicing variant of RAC1, the authors did not show this splicing event. Do IL6 and epithelial-to-cancer differentiation alter splicing variants? To determine this, mRNA levels of RAC1B and RAC1 should be also shown by conventional PCR analysis in non-polarized and polarized Caco2 cells (Fig 1C), and Caco2 cells upon coculture with macrophages and IL6 (Fig. 5 and 6). Only protein levels are not sufficient to evaluate the splicing events.

We agree with the reviewer that detection of the RAC1B transcript levels are equally important.

Thus, we included a conventional PCR image showing the qualitative mRNA levels of RAC1B and RAC1 in all cell lines of Fig. 1C, including non-polarized and polarized Caco2 cells. In addition, we moved the quantification of RAC1B transcripts in co-cultured Caco-2 cells from supplemental Fig. S3 to the main manuscript Fig. 3. Moreover, we quantified RAC1B mRNA levels in all experiments with the alternative T84 cell model, i.e. upon co-culture and IL-6 stimulation (see new Fig. 7), being equivalent to the experiments shown in Figs 5 and 6.

Please note that, as described in the methods section, the variations in RAC1B transcript levels shown in the quantitative plots express the quantification of RAC1B mRNA relative to total RAC1 mRNA in each sample, normalized to the ratio in the respective control condition sample.

We have preliminary evidence that the expression ratio of other tumour-related alternative splicing events is also affected under the described co-culture conditions; however, the full clarification of this issue will require a genome-wide approach that will be pursued in future studies.

Reviewer 3 Report

Pereria et al in their manuscript titled “Pro-inflammatory cytokines trigger the overexpression of tumour-related splice variant RAC1B in polarized colorectal cells“ by using an epithelial-like layer of polarized Caco-2 colorectal cells in co-culture with different stromal cell lines (fibroblasts, monocytes and macrophages) analysed the effect on RAC1B expression in the Caco-2 cells by RT-PCR, Western blot and confocal fluorescence microscopy and found that the presence of cancer-associated fibroblasts and M1 macrophages induced the most significant increase in RAC1B levels in the polarized colorectal cells, accompanied by a progressive loss of epithelial organization. They have also identified interleukin-6 as the main trigger for the increase in RAC1B levels, associated with STAT3 activation. In addition, they showed that IL-6 neutralization by antibodies abrogated both RAC1B overexpression and STAT3 phosphorylation in colorectal cells. The authors conclude that their results demonstrate that pro-inflammatory extracellular signals from stromal cells can trigger the overexpression of the tumour-related variant RAC1B in polarized colorectal cancer cells and their results will help to understand the tumour-promoting effect of inflammation and identify novel therapeutic strategies.

Th study was well planned, experimented were conducted diligently and presented clearly. However, the fundamental problem with this work is that they have done all the experiments only in one cell line (CaCo2) though they have used T48 cells and provided the data as supplementary material, a close look at the WB shows that there is hardly any difference. The authors should show clearly that their findings can be replicated in another cell line or conduct more experiments on T48 to stress that the mechanism is same like in CaCo2 cells in T48 as well, as the policy of the journal is that one has to show the results in at least two cell lines to be considered.

Author Response

Comment Reviewer #2: The authors should show clearly that their findings can be replicated in another cell line or conduct more experiments on T48 to stress that the mechanism is same like in CaCo2 cells in T48 as well, as the policy of the journal is that one has to show the results in at least two cell lines to be considered.

With respect to the policy of the journal that results need to be shown in at least two cell lines, we have performed more detailed experiments with the T84 colorectal cell line, which has been described to be able to differentiate into a comparable polarized epithelial-like cell layer. Because each experiment requires a 12 day-period for the cell polarization to occur, the revision of this manuscript was not possible in a shorter time period.

As a result, we can now demonstrate in this revised manuscript version that the main findings presented in the manuscript do not only apply to Caco-2, but also T84 cells. In particular, we show the following experiments:

1) More co-culture experiments were performed between polarized T84 cells and either CAFs and/or M1 macrophages to turn the data more robust.

2) We show that addition of neutralizing anti-IL-6 antibodies to the co-culture medium prevented the described increase in RAC1B.

3) Polarized T84 cells were further treated with purified IL-6 and this triggered an increase in endogenous RAC1B expression, both at the mRNA and protein levels.

4) The response of polarized T84 to IL-6 also involved an increase in pSTAT-3 levels.

Given the importance for the journal Cancers of showing our results in a second cell line model, we moved the data from the Supplementary File to the main manuscript file and joined all the above mentioned experimental data on the T84 cells into a new Figure 7. These results are described in the revised manuscript version in an additional subsection 3.4 at the end of the Result section, as detailed below.

3.4 Interleukin-6 also stimulates RAC1B expression in polarized T84 colorectal cells

In order to demonstrate that the observed effects of co-culture and IL-6 on RAC1B expression can be replicated in another colorectal cell line, we grew a polarized layer of T84 cells as a second model (Figure 7A). First, we determined the effect on RAC1B expression under the three most relevant co-culture conditions identified above: T84 with T84 (control), T84 with CT5.3, and T84 with CT5.3 plus M1. As shown in Figure 7B-C, RAC1B protein levels in polarized T84 also reached a statistical significant increase after 48 h of co-culture with CT5.3 plus M1. This increase was also observed at the respective mRNA level (Fig. 7D). The addition of a neutralizing anti-IL-6 antibody to the co-culture condition prevented the observed increase in RAC1B protein levels (Figure 7 E-F). Consistently, the addition of purified IL-6 to the basolateral side of polarized T84 cells was sufficient to induce an increase in RAC1B protein and mRNA levels (Figure 7 G-I). In all these experiments, the co-culture or IL-6-mediated effects were reflected by changes in the phosphorylation of STAT3.